# Microwave Sintering and Microwave Dielectric Properties of (1–*x*)Ca_0.61_La_0.26_TiO_3_-*x*Nd(Mg_0.5_Ti_0.5_)O_3_ Ceramics

**DOI:** 10.3390/ma14020438

**Published:** 2021-01-17

**Authors:** Shuwei Yang, Bingliang Liang, Changhong Liu, Jin Liu, Caisheng Fang, Yunlong Ai

**Affiliations:** 1Key Laboratory for Microstructural Control of Metallic Materials of Jiangxi Province, Nanchang Hangkong University, Nanchang 330063, China; 1901085204103@stu.nchu.edu.cn (S.Y.); 27014@nchu.edu.cn (C.L.); 2School of Materials Science and Engineering, Nanchang Hangkong University, Nanchang 330063, China; 1801085204026@stu.nchu.edu.cn (J.L.); 2001085600127@stu.nchu.edu.cn (C.F.); 27008@nchu.edu.cn (Y.A.)

**Keywords:** (1–*x*)CLT-*x*NMT ceramics, microwave sintering, microwave dielectric properties, lattice constant

## Abstract

The (1–*x*)Ca_0.61_La_0.26_TiO_3_-*x*Nd(Mg_0.5_Ti_0.5_)O_3_ [(1–*x*)CLT-*x*NMT, *x* = 0.35~0.60] ceramics were prepared via microwave sintering. The effects of sintering temperature and composition on the phase formation, microstructure, and microwave dielectric properties were investigated. The results show that the microwave sintering process requires a lower sintering temperature and shorter sintering time of (1–*x*)CLT-*x*NMT ceramics than conventional heating methods. All of the (1–*x*)CLT-*x*NMT ceramics possess a single perovskite structure. With the increase of *x*, the dielectric constant (*ε*) shows a downward trend; the quality factor (*Qf*) drops first and then rises significantly; the resonance frequency temperature coefficient (*τ_f_*) keeps decreasing. With excellent microwave dielectric properties (*ε* = 51.3, *Qf* = 13,852 GHz, *τ_f_* = −1.9 × 10^−6^/°C), the 0.65CLT-0.35NMT ceramic can be applied to the field of mobile communications.

## 1. Introduction

With the advent of the 5G era, microwave dielectric ceramics attract more and more attention [1]. Microwave dielectric ceramics can not only be used as insulating substrates material in microwave circuits, also as the key basic material to fabricate dielectric resonators, dielectric filters, dielectric oscillators, phase shifters, microwave capacitors, etc., for microwave communication technology [2]. Therefore, microwave components play an increasingly important role in miniaturization, integration, and cost reduction of modern communication tools [3]. The dielectric materials with high dielectric constant, high *Qf* value, near-zero temperature coefficient of resonance frequency, and low sintering temperature are strong candidates for 5G technology [4].

The Ca_0.61_La_0.26_TiO_3_ (CLT) ceramic, with typical perovskite structure, is characterized by a high dielectric constant (*ε* = 120) and a high quality factor (*Qf* = 10,700 GHz), but a very high positive resonant frequency temperature coefficient (*τ_f_* = 304 × 10^−6^/°C) [5]. The Nd(Mg_0.5_Ti_0.5_)O_3_ (NMT) ceramic also has a perovskite structure with *Qf* value of 36,900~151,000 GHz, the *ε* value is only 25~26, and the *τ_f_* is a large negative value (−72 × 10^−6^~−47 × 10^−6^/°C) [6,7]. The microwave dielectric ceramics with moderate *ε*, high *Qf*, and *τ_f_* of close to zero can be obtained, by combining Ca_0.8_Sr_0.2_TiO_3_ or CLT with NMT ceramics [8,9]. However, preparing the CLT-NMT dielectric ceramics by conventional sintering requires excessively high sintering temperature and long sintering time (1650 °C, 3 h; according to our previous work). A small amount of CuO, ZnO and other sintering aids can be added to reduce the sintering temperature [10,11], but it is difficult to avoid the introduction of the second phase and reduction of the microwave dielectric properties.

As an efficient sintering method for materials, microwave sintering can effectively reduce the sintering temperature, increase the sintering rate, and promote the grain refinement of ceramics, thus improving the microwave dielectric properties [12,13]. Up to now, there have been no reports on the preparation of the (1−*x*)CLT-*x*NMT ceramics by microwave sintering. In this work, the (1−*x*)Ca_0.61_La_0.26_TiO_3−_*x*Nd(Mg_0.5_Ti_0.5_)O_3_[(1−*x*)CLT-*x*NMT, *x* = 0.35~0.60] ceramics were prepared by microwave sintering and an in-depth study was conducted of the effects of sintering process and component ratio on its phase composition, microstructure, and microwave dielectric properties.

## 2. Materials and Methods

The (1–*x*)Ca_0.61_La_0.26_TiO_3_-*x*Nd(Mg_0.5_Ti_0.5_)O_3_ [(1–*x*)CLT-*x*NMT, *x* = 0.35~0.60] ceramics were prepared by the solid-state reaction method. The ingredients were proportioned according to the stoichiometric ratio. High-purity CaCO_3_ (99.8%, Langfang Pengcai Fine Chemical, Langfang, China), La_2_O_3_ (99.9%, Jiangxi Golden Century Advanced Materials Co., Ltd., Nanchang, China), Nd_2_O_3_ (99.9%, Jiangxi Golden Century Advanced Materials Co., Ltd., Nanchang, China), MgO (99.99%, Jiangxi Golden Century Advanced Materials Co., Ltd., Nanchang, China), and TiO_2_ (99.5%, Shanghai Jianglu Titanium Dioxide Chemical, Shanghai, China) powders were mixed by ball mill for 8 h and then dried for 24 h, ground, sieved (200 mesh), and calcined (CLT at 1200 °C for 3 h, NMT at 1400 °C for 3 h, respectively). Then, the calcined CLT and NMT powders were mixed by ball mill for 8 h, dried for 24 h, and sieved (200 mesh). After added 10 wt % of polyvinyl alcohol solution (PVA, 10%) as a binder, the mixed powders were pressed into columns with a diameter of 13 mm and a thickness of 2~6 mm and then these specimens were heated at 600 °C for 1 h to remove the PVA. Finally, these specimens were sintered in air in a microwave sintering furnace (Changsha Longtai Technology Co., Ltd., Changsha, China) (1475~1575 °C, 30 min).

The density was measured by the Archimedes method. After crushed and ground, the phase analysis of (1–*x*)CLT-*x*NMT samples was conducted by X-ray diffraction (XRD, Bruker, Bremen, Germany). After the samples were polished and cleaned with ultrasonic cleaner, etched at 50 °C lower than sintering temperature for 30 min, their microstructures were observed by a scanning electron microscope (SEM, FEI, Hillsboro, OR, USA).

To measure the dielectric properties, polished (1–*x*)CLT-*x*NMT ceramic cylindric specimen was put in a metal cavity of vector network analyzer (N5230A, Agilent Technologies, Loveland, CO, USA), in which high-frequency electromagnetic field can keep oscillating without radiation loss. The dielectric constant (*ε*) and quality factor (*Q*) were measured at 25 °C. The temperature coefficient of resonant frequency (*τ_f_*) was calculated by using the Equation (1):(1)$$\tau_{f}=\frac{f_{2}-f_{1}}{f_{1}\left(T_{2}-T_{1}\right)}$$
where *f*_1_ and *f*_2_ represent the resonant frequency at *T*_1_ (25 °C) and *T*_2_ (85 °C), respectively.

## 3. Results and Discussion

### 3.1. Sintering Characteristics

The influence of sintering temperature on the density (*ρ*) of (1−*x*)CLT-*x*NMT ceramics is shown in Figure 1. With the increase of sintering temperature (*T*), the *ρ* presents the tendency of increasing first. However, with the further increase of *T*, the *ρ* tends to decrease. It may be attributed to oversintering.

The relationship between the *ρ* and relative density (*ρ_r_*) of the (1−*x*)CLT-*x*NMT ceramic with *x* is shown in Figure 2. It can be seen intuitively that the *ρ* increases with the increase of *x*, up to 5.457 (*x* = 0.60), mainly because the density of NMT ceramic (6.16 g/cm^3^) is higher than that of CLT ceramic (4.51 g/cm^3^). The *ρ_r_* is all higher 95.5% with slightly floating and reaches 96.9% when *x* = 0.50.

### 3.2. Phase and Microstructure

The XRD patterns of the (1−*x*)CLT-*x*NMT ceramics are illustrated in Figure 3. The diffraction peak positions are almost completely overlapped in the composition range of *x* = 0.35~0.60, indicating a perovskite structure without second phase. It should be pointed out that superlattice diffraction peaks were observed when *x* = 0.40 and 0.45. The enlarged part of 32.1~33.3°, as shown in the upper right corner of Figure 3, indicates that the main diffraction peaks of (1−*x*)CLT-*x*NMT ceramics shift toward low angle with the increase of *x*. It suggests the increasing lattice constant of the identified perovskite structure.

The lattice constant (*a*, *b*, *c*) and unit cell volume (*V_u_*) of (1−*x*)CLT-*x*NMT ceramics are shown in Figure 4. Both lattice constant and unit cell volume gradually increase with the increasing *x*, which is in accordance with the XRD analysis. This trend depends on two factors: the decreasing vacancy concentration in *A*-site, the increasing Mg^2+^ content (*r*(Mg^2+^) > *r*(Ti^4+^), *r*(Mg^2+^) = 0.072 nm, *r*(Ti^4+^) = 0.061 nm when CN = 6) in *B*-site [14], with the increase of NMT content in (1−*x*)CLT-*x*NMT ceramics.

SEM images of the (1–*x*)CLT-*x*NMT ceramics (1550 °C, 30 min) are presented in Figure 5. When *x* ≤ 0.55, the grain size (10~30 μm) is relatively uniform and change slightly with the increase of NMT content. When *x* = 0.60, the grain size (20~50 μm) is significantly larger than that of the rest composition. When *x* < 0.60, strip-shaped grains can be observed, which is similar to the CaTiO_3_-La(Mg_0.5_Ti_0.5_) ceramics [15].

### 3.3. Microwave Dielectric Properties

The relationship between dielectric constant (*ε*) and composition of the (1–*x*)CLT-*x*NMT ceramics is illustrated in Figure 6. With the increase of *x*, the *ε* gradually decreases from 51.3 to 36.4 because the *ε* of NMT (~24) is much lower than that of CLT (~120). To evaluate the influence of porosity (*p*) on the *ε*, the theoretical dielectric constant (*ε_th_*) of (1–*x*)CLT-*x*NMT ceramics can be calculated according to the following equation [16,17]:(2)$$\varepsilon_{th}=\varepsilon /(1-\frac{3p(\varepsilon -1)}{2\varepsilon +1})$$
where *ε_th_* is the dielectric constant of a theoretically fully dense ceramic, *ε* is the measured dielectric constant, *p* is the porosity (*p* = 100%–*ρ_r_*). Furthermore, Equation (2) can be simplified as follows due to *ε* >> 1:(3)$$\varepsilon_{th}=\frac{\varepsilon }{1-1.5p}$$

As shown in Figure 6, the *ε_th_* of (1–*x*)CLT-*x*NMT ceramics decreases from 54.2 to 38.3 with the increase of *x*. It indicates an improvement space of 4.9~6.4%.

The *Qf* value of the (1−*x*)CLT-*x*NMT ceramics is presented in Figure 7. It ascends from 13,852 GHz (*x* = 0.35) to 17,148 GHz (*x* = 0.40) and then drops to 8482 GHz (*x* = 0.45) and finally climbs to 32,637 GHz (*x* = 0.60). Generally, the appearance of superlattice diffraction peaks is related to the 1:1 ordering of Mg^2+^ and Ti^4+^ [15], which often affects the dielectric loss and then *Qf*. The dielectric loss decreases with increasing of ions’ degree of order, but increases with attenuation of ions’ phonon mode. As *x* increases to 0.40, the ions’ degree of order constantly deepens and the phonon mode attenuates slightly, which results in an increase of *Qf*. When *x* climbs to 0.45, the ions’ degree of order continues to deepen, but the phonon mode attenuates intensively, which leads to a decrease in *Qf*. Later, the further increase of *x* transforms the (1−*x*)CLT-*x*NMT ceramics from a CLT-based ordered solid solution to an NMT-based ordered solid solution, decreasing dielectric loss, and increasing the *Qf* value to 32,637 GHz (*x* = 0.60).

Similarly, the effect of porosity (*p*) on the *Qf* value (with 10^3^~10^4^ GHz order of magnitude) can be evaluated by the following equation [18]:(4)$$Q=Q_{0}(1-1.5p)$$
where *Q*_0_ is the intrinsic quality factor, and *p* is the porosity. The results suggest that an improvement space of 503~1741 GHz.

The temperature coefficient of resonance frequency (*τ_f_*) and tolerance factor (*t*) of the (1−*x*)CLT-*x*NMT ceramics are shown in Figure 8. The relationship between the *τ_f_* and temperature coefficient of dielectric constant (*τ_ε_*) and linear expansion coefficient (*α_L_*) can be identified as follows [19,20]:(5)$$\tau_{f}=-\frac{1}{2}\tau_{\varepsilon }-\alpha_{L}$$
where the *α_L_* of ceramics is 6~10 × 10^−6^/°C [21]. Therefore, the value of the *τ_f_* depends on the *τ_ε_*.

In 1926, Goldschmidt [22] initially proposed the tolerance factor (*t*) to evaluate the stability of crystal structure. As to perovskite structure (ABO_3_), the *t* can be calculated according to the following equation [23]:(6)$$t=\frac{R_{A}+R_{O}}{\sqrt{2}(R_{B}+R_{O})}$$
where *R_A_*_,_
*R_B_* and *R_O_* are the radius of *A*-site ions, *B*-site ions and O^2−^, respectively. The effective ionic radius from Shannon [14] were used to calculate the *t* of (1−*x*)CLT-*x*NMT ceramics. Generally, the *t* of the perovskite structure should be in the range of 0.77~1.1 and the closer to 1 *t* is, the stabler the perovskite structure is.

Colla et al. [21] studied the relationship between the tilt of BO_6_ octahedron in ABO_3_-type perovskite lattice and the temperature coefficient of dielectric constant (*τ**_ε_*). The results show that the *τ**_ε_* is mainly affected by the tilt of BO_6_ octahedron. The increasing tilt of BO_6_ octahedron will result in the change of the *τ**_ε_* to the positive direction. The tilt degree of BO_6_ octahedron can be described by the *t*: greater difference between the *t* value and 1 means greater tilt degree [23]. The *τ**_ε_* and the *t* have the following regularity: when *t* < 0.965, the decrease of *t* will lead to the change of *τ**_ε_* to the positive direction. For the (1−*x*)CLT-*x*NMT ceramics, the decrease of *t* value will lead to the increasing tilt degree of the BO_6_ octahedron. Thus the *τ**_ε_* will increase and the *τ_f_* will decrease to the negative direction, as shown in Figure 8.

Our previous work showed that the conventional sintering procedure of (1–*x*)CLT-*x*NMT ceramics was sintering at 1650 °C for 3 h. In the present work, microwave sintering processes effectively promoted the densification of (1–*x*)CLT-*x*NMT ceramics with lower sintering temperature (1475 °C) and shorter sintering time (30 min). Chen et al. [9] prepared 0.40Nd(Mg_1/2_Ti_1/2_)O_3_-0.60Ca_0.6_La_0.8/3_TiO_3_ (with 1 wt % B_2_O_3_ as sintering additive) ceramics via conventional sintering (1375 °C, 3 h) with excellent microwave dielectric properties: *ε* = 49, *Qf* = 13,000 GHz, *τ_f_ =* 1 × 10^−6^/°C. As compared to Chen’s work, microwave sintered 0.65CLT-0.35NMT ceramics (without sintering additive, 1550 °C, 30 min) also possesses similar microwave dielectric properties: *ε* = 51.3, *Qf* = 13,852 GHz, *τ_f_* = −1.9 × 10^−6^/°C.

## 4. Conclusions

The (1–*x*)Ca_0.61_La_0.26_TiO_3_-*x*Nd(Mg_0.5_Ti_0.5_)O_3_[*x* = 0.35~0.60, (1–*x*)CLT-*x*NMT] ceramics were prepared by microwave sintering. The effects of sintering process and component distribution compare on its phase composition, microstructure, and microwave dielectric properties were investigated. Microwave sintering can effectively reduce the sintering temperature and the sintering time. The (1–*x*)CLT-*x*NMT ceramics have formed a perovskite structure. As *x* increases, the *ε* shows a downward trend, the *Qf* first drops to 8482 GHz and then rises to 32,637 GHz, and the *τ_f_* keeps decreasing. When *x* = 0.35, the comprehensive microwave dielectric performance is: *ε* = 51.3, *Qf* = 13,852 GHz, *τ_f_* = −1.9 × 10^−6^/°C (1550 °C, 30 min). The (1–*x*)CLT-*x*NMT ceramics can be applied to the field of mobile communications.

## Figures and Tables

**Figure 1 materials-14-00438-f001:**
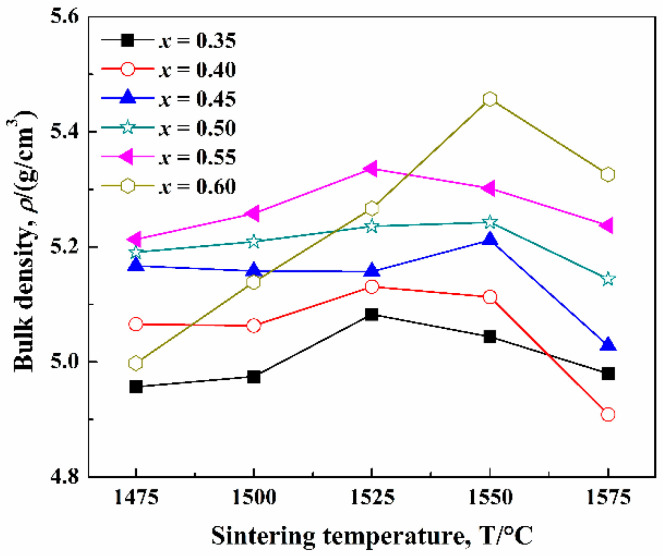
Bulk density of the (1–*x*)CLT-*x*NMT ceramics.

**Figure 2 materials-14-00438-f002:**
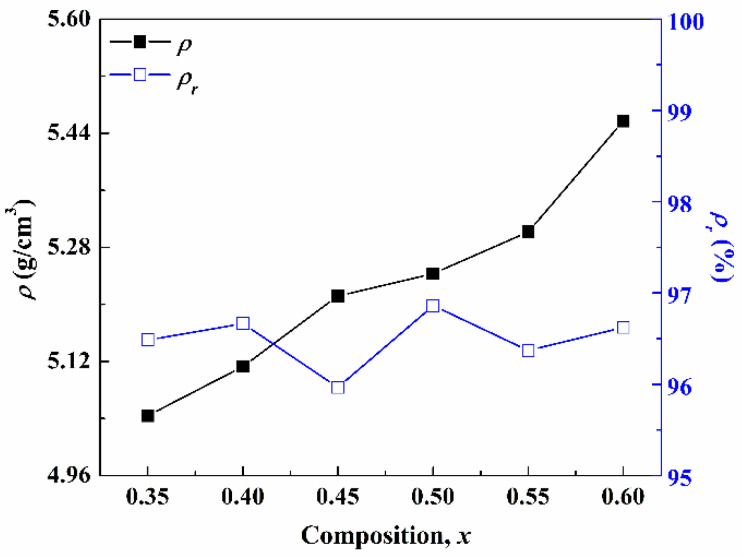
The curves of bulk density and relative density with the composition of (1–*x*)CLT-*x*NMT ceramics.

**Figure 3 materials-14-00438-f003:**
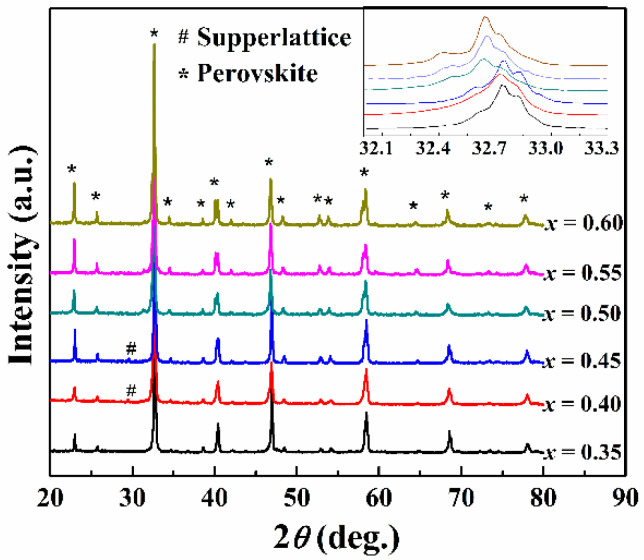
XRD patterns of the (1–*x*)CLT-*x*NMT ceramics.

**Figure 4 materials-14-00438-f004:**
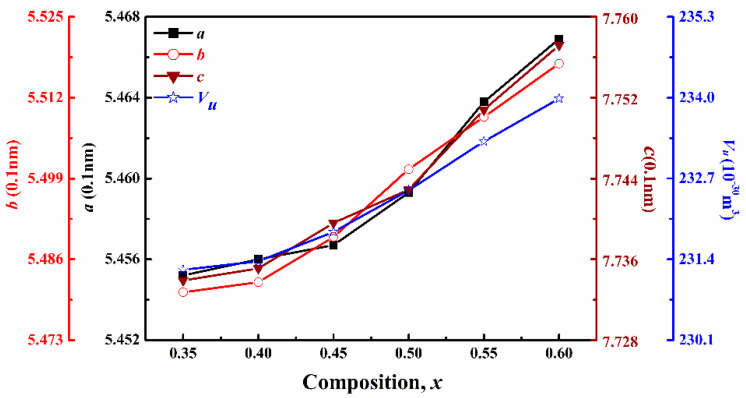
Lattice constant and unit cell volume of the (1–*x*)CLT − *x*NMT ceramics.

**Figure 5 materials-14-00438-f005:**
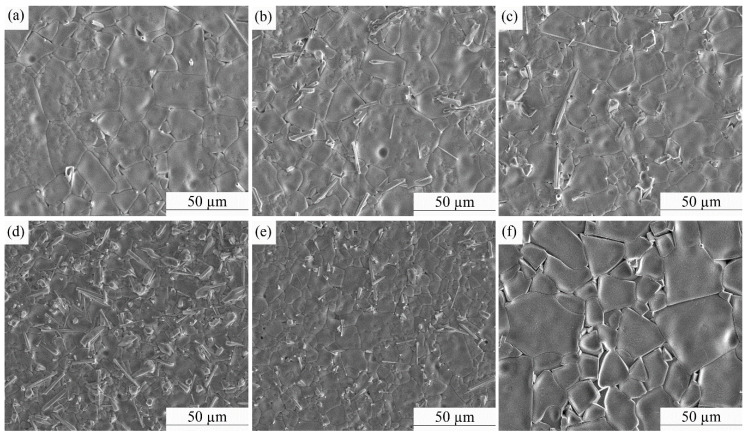
SEM images of the (1–*x*)CLT-*x*NMT ceramics (1550 °C, 30 min): (**a**) *x* = 0.35, (**b**) *x* = 0.40, (**c**) *x* = 0.45, (**d**) *x* = 0.50, (**e**) *x* = 0.55, (**f**) *x* = 0.60.

**Figure 6 materials-14-00438-f006:**
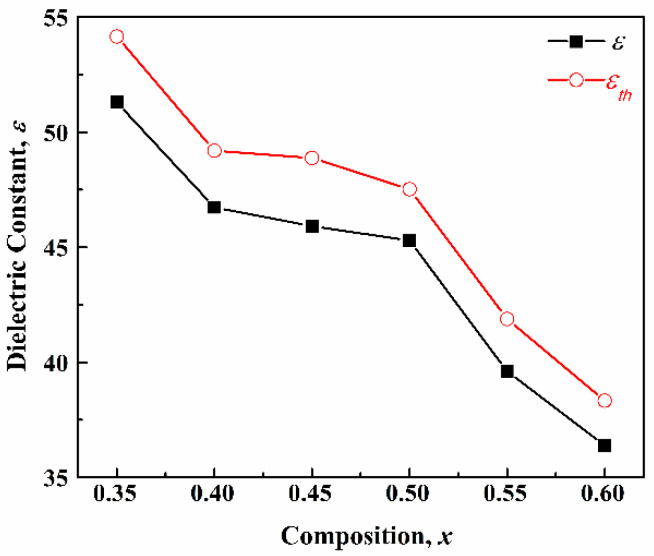
*ε* and *ε_th_* of the (1–*x*)CLT-*x*NMT ceramics.

**Figure 7 materials-14-00438-f007:**
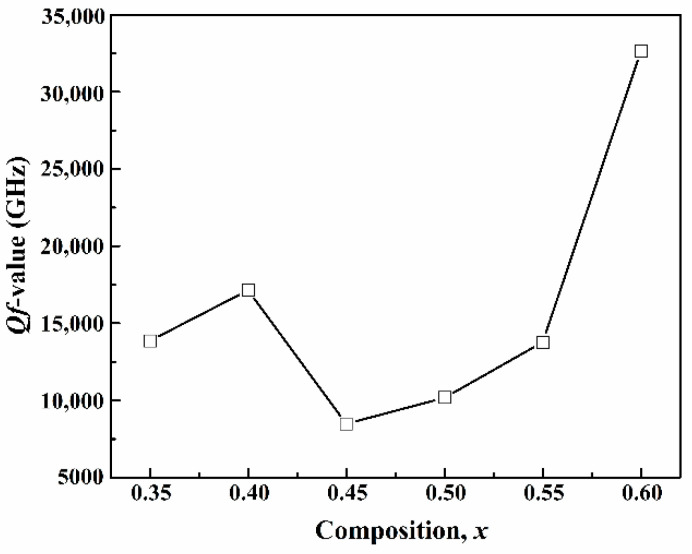
The *Qf* value of the (1–*x*)CLT-*x*NMT ceramics.

**Figure 8 materials-14-00438-f008:**
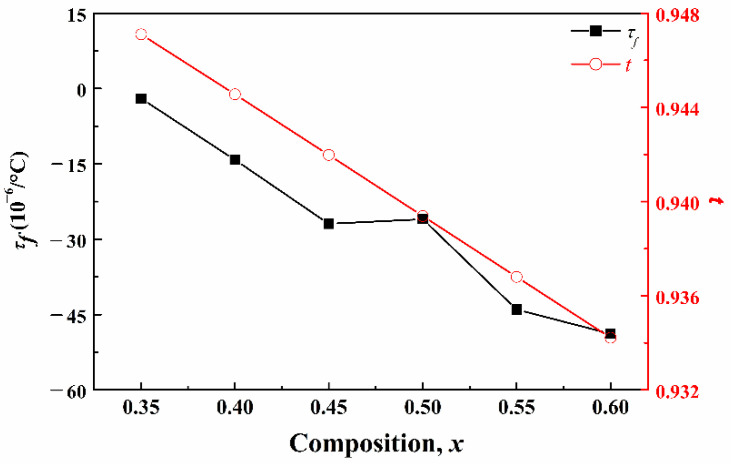
*τ_f_* and *t* of the (1–*x*)CLT-*x*NMT ceramics.

## Data Availability

The data presented in this study are available on request from the corresponding author. The data are not publicly available due to privacy.

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
