# Peer review of "Microwave Sintering and Microwave Dielectric Properties of (1–x)Ca0.61La0.26TiO3-xNd(Mg0.5Ti0.5)O3 Ceramics"

_materials, 2021, doi:10.3390/ma14020438_

Round 1
Reviewer 1 Report
In the present manuscript, “Microwave Sintering and Microwave Dielectric Properties of (1–x)Ca0.61La0.26TiO3-xNd(Mg0.5Ti0.5)O3 Ceramics” authors studied the microwave sintering effect on microwave dielectric properties. However, there are a few points that need to be addressed. I didn’t recommend this manuscript in the present form for publication due to the following reasons:
- In XRD, the samples x =0.40 and x= 0.45 have a superlattice plane; please explain the reason for its formation and its effect on microwave dielectric properties.
- How were the relative densities calculated for (1-x) CLT-xNMT compositions? Have you calculated the theoretical densities of each sample?
- Authors not discussed how the microwave sintering affects the microwave dielectric properties. If the microwave sintering is shown better properties, then it is important to compare with conventional sintering results.
- The Q of 40CLT-60NMT composition is shown an optimum value, then why the authors did not study the above concentration.
Author Response
Ref. No.: Materials-998994
Title: Microwave Sintering and Microwave Dielectric Properties of (1–x)Ca0.61La0.26TiO3-xNd(Mg0.5Ti0.5)O3 Ceramics Thanks to the reviewers for your time and thoughtful comments, which have been incorporated into the revised manuscript. Hopefully we have addressed all of your concerns.
Our responses to the Reviewer’s comments are presented in BOLD type as follows. The page and line numbers refer to our revised manuscript submitted at 12/20/2020.
Reviewer #1: In the present manuscript, “Microwave Sintering and Microwave Dielectric Properties of (1–x)Ca0.61La0.26TiO3- xNd(Mg0.5Ti0.5)O3 Ceramics” authors studied the microwave sintering effect on microwave dielectric properties. However, there are a few points that need to be addressed. I didn’t recommend this manuscript in the present form for publication due to the following reasons:
- In XRD, the samples x=0.40 and x=0.45 have a superlattice plane; please explain the reason for its formation and its effect on microwave dielectric properties. Response: Generally, the appearance of superlattice diffraction peaks is related to the 1:1 ordering of Mg2+ and Ti4+[*], which often affects the dielectric loss and then Qf. The dielectric loss decreases with increasing of ions’ degree of order, but increases with attenuation of ions’ phonon mode. As x increases to 0.40, the ions’ degree of order constantly deepens and the phonon mode attenuates slightly, which results in an increase of Qf. When x climbs to 0.45, the ions’ degree of order continues to deepen, but the phonon mode attenuates intensively, which leads to a decrease in Qf. Later, the further increase of x transforms the (1−x)CLT-xNMT ceramics from a CLT-based ordered solid solution to an NMT-based ordered solid solution, decreasing dielectric loss and increasing the Qf value (in RED font, Line 136-144). [*] Kipkoech E R, Azough F, Freer R. Microstructural control of microwave dielectric properties in CaTiO3-La(Mg1/2Ti1/2)O3 ceramics[J]. J Appl Phys, 2005, 97(064103): 1-11.
- How were the relative densities calculated for (1–x)CLT-xNMT compositions? Have you calculated the theoretical densities of each sample? Response: First, the lattice parameters (a, b, c and v) of (1–x)CLT-xNMT ceramics with orthorhombic structure were calculated on the basis of XRD analysis. Second, the theoretical density (ρth, g/cm3) was calculated according to the following equation: where Mr (g/mol) is relative molecular mass of (1–x)CLT-xNMT, z is the number of (1–x)CLT-xNMT contained in a unit cell, v is cell volume, NA is Avogadro constant (6.02214076×10²³). The lattice parameters and relative densities of (1–x)CLT-xNMT ceramics were listed in the following table: Table 1. Lattice parameters and theoretical density of (1–x)CLT-xNMT ceramics. x 0.35 0.40 0.45 0.50 0.55 0.60 Mr(g/mol) 181.5918 185.1866 188.7814 192.3762 195.971 199.5658 a(10-10m) 5.4552 5.456 5.4567 5.4593 5.4638 5.4669 b(10-10m) 5.4807 5.4823 5.4896 5.5005 5.5089 5.5175 c(10-10m) 7.7339 7.7351 7.7396 7.7429 7.7508 7.7572 Vu(10-30m3) 231.2306 231.3679 231.8405 232.5106 233.2954 233.9852 z 4 4 4 4 4 4 ρth(g/cm3) 5.216 5.316 5.409 5.496 5.579 5.665 Third, the relative densities (r, %) of (1–x)CLT-xNMT ceramics were calculated according to the following equation: where ρ is measured density (g/cm3).
- Authors not discussed how the microwave sintering affects the microwave dielectric properties. If the microwave sintering is shown better properties, then it is important to compare with conventional sintering results. Response: Our previous work showed that the conventional sintering procedure of (1–x)CLT-xNMT ceramics was sintering at 1650 °C for 3h. In the present work, microwave sintering processes effectively promoted the densification of (1–x)CLT-xNMT ceramics with lower sintering temperature (1475 °C) and shorter sintering time (30 min). Chen[*] prepared 0.40Nd(Mg1/2Ti1/2)O3- 0.60Ca0.6La0.8/3TiO3 (with 1wt% B2O3 as sintering additive) ceramics via conventional sintering (1375 °C, 3h) with excellent microwave dielectric properties: ε=49, Qf=13000GHz, τf =1×10−6/°C. As compared to Chen’s work, microwave sintered 0.65CLT-0.35NMT ceramics (without sintering additive, 1550 °C, 30 min) also possesses similar microwave dielectric properties: ε=51.3, Qf=13852GHz, τf=−1.9×10−6/°C (in RED font, Line 174~182). [*] Chen, Y.B. Structure and dielectric characterization of xNd(Mg1/2Ti1/2)O3- (1−x)Ca0.6La0.8/3TiO3 in the microwave frequency range. J. Alloys Compd. 2011, 509: 6285-6288, doi: 10.1016/j.jallcom.2011.03.056
- The Q of 0.40CLT-0.60NMT composition is shown an optimum value, then why the authors did not study the above concentration. Response: In order to prevent the center frequency of microwave devices from drifting significantly with the change in temperature, f should be adjusted to near zero. In present work, the f of 0.65CLT-0.35NMT ceramic is close to zero (−1.9×10−6/°C) and which of 0.40CLT-0.60NMT ceramic is −48×10−6/°C. Meanwhile, we discussed the changing tendency of Qf with x. (in RED font, Line 136~144).

Reviewer 2 Report
In this work the authors have presented the results of the effects of sintering temperature and composition on the phase formation, microstructure and microwave dielectric properties for (1–x)Ca0.61La0.26TiO3-xNd(Mg0.5Ti0.5)O3[(1–x)CLT-xNMT, x=0.35~0.60] ceramics. Four different ratios were investigated.
General remarks
The topic is interesting and consistent with the contents to be proposed to the readers of the MDPI Materials Journal. I suggest to the Authors to take into account the following revisions, which I feel should be addressed and clarified:
- In Abstract could just be explained what is investigated and what is concluded, with less experimental data
- Please update Introduction with up-to-date literature.
- 42nd row – “…time is too longer”? What does that mean?
- Materials and methods – For chemicals, please name the suppliers Also, please provide details aboe used techniques.
- Also in Materials and methods – XRD is Bruker I suppose, not Brooke
- In 105th row, r(Nd3+)r(Ti4+) (where r(Nd3+)=0.127 nm, r(Ca2+ 105 )=0.134 nm, r(Mg2+)=0.072 nm, r(Ti4+)=0.061 nm) – Maybe compose a sentence with further explanation of those radii?
- Figure 6. Why ε of the (1–x)CLT-xNMT is increasing for x=0.4, and for all other ratios is decreasing? Do you have explanation for that? It could be interesting.
- The porosity has a great influence on dielectric loss. Generally, the effect of porosity can be divided into two categories: materials with smaller Qf value (103 GHz order of magnitude). 171st row – why second category is not defined?
- How were dielectric measurements performed? Could further explanation for experimental setup be provided? Shown results are for what temperature? It is mentioned that measurements were performed at 25 and 75 °C, but for which of those are discussed results?
- Please check once again spelling and big/small letters (for example, there is “_“ in the middle of sentence -192nd row).
- Please check language style and grammar.
In light of it, my suggestion is revision before further consideration for publication.
I ask the Editor to give me the possibility to see the reviewed manuscript.
Author Response
Ref. No.: Materials-998994
Title: Microwave Sintering and Microwave Dielectric Properties of (1–x)Ca0.61La0.26TiO3-xNd(Mg0.5Ti0.5)O3 Ceramics Thanks to the reviewers for your time and thoughtful comments, which have been incorporated into the revised manuscript. Hopefully we have addressed all of your concerns.
Our responses to the Reviewer’s comments are presented in BOLD type as follows. The page and line numbers refer to our revised manuscript submitted at 12/20/2020.
Reviewer #2:
In this work the authors have presented the results of the effects of sintering temperature and composition on the phase formation, microstructure and microwave dielectric properties for (1–x)Ca0.61La0.26TiO3-xNd(Mg0.5Ti0.5)O3 [(1–x)CLT-xNMT, x=0.35~0.60] ceramics. Four different ratios were investigated. The topic is interesting and consistent with the contents to be proposed to the readers of the MDPI Materials Journal. I suggest to the Authors to take into account the following revisions, which I feel should be addressed and clarified:
- In Abstract could just be explained what is investigated and what is concluded, with less experimental data. Response: In Abstract, some experimental data were deleted and the revised part was marked in RED color (Line 19~22).
- Please update Introduction with up-to-date literature. Response: We updated Introduction and References with up-to-date literature (in RED font, Line 27~45).
- 42nd row – “…time is too longer”? What does that mean? Response: We revised our expression to “long sintering time (3h)” (in RED font, Line 42).
- Materials and methods–For chemicals, please name the suppliers. Also, please provide details about used techniques. Response: We have added the suppliers of the chemicals and details of used techniques to “Materials and Methods” (in RED font, Line 56~62).
- Also in Materials and methods – XRD is Bruker I suppose, not Brooke Response: We’d like to extend our sincere gratitude to the reviewer for his pointing out this mistake. We have corrected “Brooke” as “Bruker” (in RED font, Line 68).
- In 105th row, r(Nd3+)r(Ti4+) (where r(Nd3+)=0.127 nm, r(Ca2+)=0.134 nm, r(Mg2+)=0.072 nm, r(Ti4+)=0.061 nm)–Maybe compose a sentence with further explanation of those radii? Response: We added the coordination number (CN) of ions at B-site (in RED font, Line 107,108)
- Figure 6. Why ε of the (1–x)CLT-xNMT is increasing for x=0.4, and for all other ratios is decreasing? Do you have explanation for that? It could be interesting. Response: We are so sorry that we made a mistake. The ε presented in Figure 6 from the original manuscript was measured at 200 kHz instead of microwave frequency. Therefore, the Figure 6 from the original manuscript and explanation about it were deleted in the revised manuscript.
- The porosity has a great influence on dielectric loss. Generally, the effect of porosity can be divided into two categories: materials with smaller Qf value (103~104 GHz order of magnitude). 171st row–why second category is not defined? Response: The second category is beyond the scope of Qf value involved in this work, so we deleted it and updated the original sentence (in RED font, Line 145~149):
- How were dielectric measurements performed? Could further explanation for experimental setup be provided? Shown results are for what temperature? It is mentioned that measurements were performed at 25 and 85 °C, but for which of those are discussed results? Response: We described the details of the dielectric measurements (in RED font, Line 72~78).
- Please check once again spelling and big/small letters (for example, there is “_“ in the middle of sentence -192nd row). Response: We have checked the whole manuscript and corrected some spelling and big/small letters mistakes (with a GREEN background).
- Please check language style and grammar. Response: This manuscript has been revised carefully by Zhiping Li, who is a teacher of Institute of Foreign Languages in Jiangxi Science and Technology Normal University and a Ph.D. candidate for English Language & Literature at University of Wales Trinity Saint David (2019-2022). We corrected some grammatical errors in the revised manuscript (with a YELLOW background).

Reviewer 3 Report
The authors systematically investigated the CLT-xNMT, composition dependence on the phase formation, microstructure and microwave dielectric properties, I have a few comments on this work as follows:
- "The microwave dielectric properties of these samples were measured by the metal cavity method [10]" The authors should give some information about sample preparation for microwave dielectric measurements. What was the measuring frequency for dielectric permittivity?
- The XRD patterns in Figure 3 should be indexed. Can the authors explain what do they mean by superlattice reflection peaks. What is their effect on perovskite phase purity?
- Concerning Eqs 2 and 3 and Figure 7. It is not clear form the manuscript how the authors determined the porosity of the sample.
- Section between lines 161-168. The presented argument does not relate to the main topic of the work. The authors did not perform IR measurements on their samples and cannot draw such conclusions. This paragraph should be deleted.
- Lines 172-174. The authors should decide how do they define p parameter. In eq. 3 p was defined as porosity. Moreover, equations 3 and 4 are essentially the same. Thus the authors repeat themselves in sections 161-168 and 172-174.
- It is unclear how the authors determined the linear expansion coefficient (αL). Why they assumed that this coefficiet is the same for all ceramics under invesitigations.
- Line 207. Please correct the definitions of the parameters dij and b.
- In order to be able to draw conclusions about the BO6 octahedron tilts, it would be necessary to do neutron or electron-diffraction measurements. Since the authors do not provide with accurate structural studies nor refer to such results, the section between lines 211 and 220 present only speculations.
- Please check the references very carefully since e.g. position 14 has wrong DOI.
Author Response
Ref. No.: Materials-998994
Title: Microwave Sintering and Microwave Dielectric Properties of (1–x)Ca0.61La0.26TiO3-xNd(Mg0.5Ti0.5)O3 Ceramics
Thanks to the reviewers for your time and thoughtful comments, which have been incorporated into the revised manuscript. Hopefully we have addressed all of your concerns.
Our responses to the Reviewer’s comments are presented in BOLD type as follows. The page and line numbers refer to our revised manuscript submitted at 12/25/2020.
Reviewer#3: The authors systematically investigated the (1–x)CLT-xNMT, composition dependence on the phase formation, microstructure and microwave dielectric properties, I have a few comments on this work as follows:
1. "The microwave dielectric properties of these samples were measured by the metal cavity method [10]" The authors should give some information about sample preparation for microwave dielectric measurements. What was the measuring frequency for dielectric permittivity?
Response:
We described the details of the dielectric measurements as follows (in RED font, Line 72~78 ):
To measure the dielectric properties, polished (1–x)CLT-xNMT ceramic cylindric specimen was put in a metal cavity of vector network analyzer (AMBIOX TECHNOIOG, N5230A), in which high-frequency electromagnetic field can keep oscillating without radiation loss.
The dielectric constant (ε) and quality factor (Q) were measured at 25 °C. The temperature coefficient of resonant frequency (τf) was calculated using the following equation:
Where f1 and f2 represent the resonant frequency at T1(25 °C) and T2(85 °C), respectively.
The measuring frequency for dielectric permittivity was from 4.10 GHz to 4.99GHz, as listed in the following Table:
Table 1. The microwave dielectric properties of (1–x)CLT-xNMT ceramics
x Sintering process f (GHz) Qf (GHz) f(10−6/°C)
0.35 1550°C, 30min 51.30 4.104 13852 -1.97
0.40 1525°C, 30min 46.74 4.204 17148 -14.19
0.45 1550°C, 30min 45.92 4.606 8482 -26.2
0.50 1525°C, 30min 45.29 4.602 10183 -26
0.55 1550°C, 30min 39.61 4.986 13773 -44
0.60 1525°C, 30min 36.39 4.755 32637 -48.81
2. The XRD patterns in Figure 3 should be indexed. Can the authors explain what do they mean by superlattice reflection peaks. What is their effect on perovskite phase purity?
Response:
Generally, the appearance of superlattice diffraction peaks is related to the 1:1 ordering of Mg2+ and Ti4+[*], which often affects the dielectric loss and then Qf. The dielectric loss decreases with increasing of ions’ degree of order, but increases with attenuation of ions’ phonon mode. As x increases to 0.40, the ions’ degree of order constantly deepens and the phonon mode attenuates slightly, which results in an increase of Qf. When x climbs to 0.45, the ions’ degree of order continues to deepen, but the phonon mode attenuates intensively, which leads to a decrease in Qf. Later, the further increase of x transforms the (1−x)CLT-xNMT ceramics from a CLT-based ordered solid solution to an NMT-based ordered solid solution, decreasing dielectric loss and increasing the Qf value to 32637 GHz (x=0.60) (in RED font, Line 136~144).
[*] Kipkoech E R, Azough F, Freer R. Microstructural control of microwave dielectric properties in CaTiO3-La(Mg1/2Ti1/2)O3 ceramics[J]. J Appl Phys, 2005, 97(064103): 1-11.
3. Concerning Eqs 2 and 3 and Figure 7. It is not clear form the manuscript how the authors determined the porosity of the sample.
Response:
First, the lattice parameters (a, b, c and v) of (1–x)CLT-xNMT ceramics with orthorhombic structure were calculated on the basis of XRD analysis.
Second, the theoretical density (ρth, g/cm3) was calculated according to the following equation:
where Mr (g/mol) is relative molecular mass of (1–x)CLT-xNMT, z is the number of (1–x)CLT-xNMT contained in a unit cell, v is cell volume, NA is Avogadro constant (6.02214076×10²³).
The lattice parameters and relative densities of (1–x)CLT-xNMT ceramics were listed in the following table:
Table 1. Lattice parameters and theoretical density of (1–x)CLT-xNMT ceramics.
x 0.35 0.40 0.45 0.50 0.55 0.60
Mr(g/mol) 181.5918 185.1866 188.7814 192.3762 195.971 199.5658
a(10-10m) 5.4552 5.456 5.4567 5.4593 5.4638 5.4669
b(10-10m) 5.4807 5.4823 5.4896 5.5005 5.5089 5.5175
c(10-10m) 7.7339 7.7351 7.7396 7.7429 7.7508 7.7572
Vu(10-30m3) 231.2306 231.3679 231.8405 232.5106 233.2954 233.9852
z 4 4 4 4 4 4
ρth(g/cm3) 5.216 5.316 5.409 5.496 5.579 5.665
Third, the relative densities (r, %) of (1–x)CLT-xNMT ceramics were calculated according to the following equation:
where ρ is measured density (g/cm3).
4. Section between lines 161-168. The presented argument does not relate to the main topic of the work. The authors did not perform IR measurements on their samples and cannot draw such conclusions. This paragraph should be deleted.
Response:
Line 161~168 of initial manuscript were deleted.
5. Lines 172-174. The authors should decide how do they define p parameter. In eq. 3 p was defined as porosity. Moreover, equations 3 and 4 are essentially the same. Thus the authors repeat themselves in sections 161-168 and 172-174.
Response:
p is the porosity (p=100%–r) of (1–x)CLT-xNMT ceramics (in RED font, Line 127).
Line 161~168 of initial manuscript were deleted.
6. It is unclear how the authors determined the linear expansion coefficient (αL). Why they assumed that this coefficient is the same for all ceramics under investigations.
Response: As to (1–x)CLT-xNMT ceramics, the influences of composition on the αL should be very slightly and we cited the conclusion of Colla et al. [*]: The αL of ceramics is generally 6~10×10−6/°C.
[*] Colla E L, Reaney I M, Setter N. Effect of structural changes in complex perovskites on the temperature coefficient of the relative permittivity[J]. J Appl Phys, 1993, 74(5): 3414-3425.
7. Line 207. Please correct the definitions of the parameters dij and b.
Response: We deleted the content about “bond valence parameter” in the initial manuscript.
8. In order to be able to draw conclusions about the BO6 octahedron tilts, it would be necessary to do neutron or electron-diffraction measurements. Since the authors do not provide with accurate structural studies nor refer to such results, the section between lines 211 and 220 present only speculations.
Response:
Just as mentioned by the reviewer, neutron or electron-diffraction measurement can evaluate accurately the BO6 octahedron tilt.
Indeed, the tolerance factor (t) is calculated on the basis of the ionic radius of A-cite ions (Ca2+, La3+, Nd3+), B-cite ions (Mg2+, Ti4+) and O2–. In this work, we introduced the tolerance factor (t) of ABO3-type perovskite structure to qualitatively describe the BO6 octahedron tilt, according to the research result of Ian M. Reaney et al. [*] (Line 165~173).
[*]Ian M. Reaney, Enrico L. Colla Enrico L. Colla and Nava Setter Nava Setter. Dielectric and Structural Characteristics of Ba- and Sr-based Complex Perovskites as a Function of Tolerance Factor. Japanese Journal of Applied Physics, Volume 33, Number 7R: 3984
9. Please check the references very carefully since e.g. position 14 has wrong DOI.
Response: We checked the references carefully.

Round 2
Reviewer 2 Report
The Authors addressed the issues raised.
Thank you.
Reviewer 3 Report
The authors presented improved version of the publication with additional data. The presented data are valuable and may be published.
Nevertheless, the new version could contain both tables presented in the authors' replies.
The other remark. I wouldn't draw too far-reaching conclusions about the relationship between the tolerance factor and oxygen octahedra tilts. The Goldschmidt tolerance factor only informs about the geometric conditions about the perovskite structure stability and how the unit cell my be deformed. It is better to use the new improved tolerance factor which takes into account the oxidation state of the A ion. But to be able to infer the type of deformation is necessary to perform structural studies.